# Effects of the Combination of the *C1473G* Mutation in the *Tph2* Gene and *Lethal Yellow* Mutations in the *Raly-Agouti* Locus on Behavior, Brain 5-HT and Melanocortin Systems in Mice

**DOI:** 10.3390/biom13060963

**Published:** 2023-06-08

**Authors:** Polyna D. Komleva, Ghofran Alhalabi, Arseniy E. Izyurov, Nikita V. Khotskin, Alexander V. Kulikov

**Affiliations:** 1Department of Psychoneuropharmacology, Federal Research Center Institute of Cytology and Genetic Siberian Branch of Russian Academy of Sciences, 630090 Novosibirsk, Russia; 2Department of Genetic Collections of Neural Disorders, Federal Research Center Institute of Cytology and Genetic Siberian Branch of Russian Academy of Sciences, 630090 Novosibirsk, Russia; 3Department of Genetics of Industrial Microorganisms, Federal Research Center Institute of Cytology and Genetic Siberian Branch of Russian Academy of Sciences, 630090 Novosibirsk, Russia

**Keywords:** serotonin system, melanocortin system, tryptophan hydroxylase 2, agouti, C1473G polymorphism, lethal yellow, behavior, brain, mice

## Abstract

Tryptophan hydroxylase 2 (TPH2) is the key and rate-limited enzyme of serotonin (5-HT) synthesis in the brain. The *C1473G* mutation in the *Tph2* gene results in a two-fold decrease in enzyme activity in the mouse brain. The *lethal yellow* (*A^Y^*) mutation in the *Raly-Agouti* locus results in the overexpression of the *Agouti* gene in the brain and causes obesity and depressive-like behavior in mice. Herein, the possible influences of these mutations and their combination on body mass, behavior, brain 5-HT and melanocortin systems in mice of the B6-1473CC/aa. B6-1473CC/*A^Y^a*, B6-1473GG/aa are investigated. B6-1473GG/*A^Y^a* genotypes were studied. The *1473G* and *A^Y^* alleles increase the activity of TPH2 and the expression of the *Agouti* gene, respectively, but they do not alter 5-HT and 5-HIAA levels or the expression of the genes *Tph2*, *Maoa*, *Slc6a4*, *Htr1a*, *Htr2a*, *Mc3r* and *Mc4r* in the brain. The *1473G* allele attenuates weight gain and depressive-like immobility in the forced swim test, while the *A^Y^* allele increases body weight gain and depressive-like immobility. The combination of these alleles results in hind limb dystonia in the B6-1473GG/*A^Y^a* mice. This is the first evidence for the interaction between the *C1473G* and *A^Y^* mutations.

## 1. Introduction

The enzyme tryptophan hydroxylase 2 (TPH2) hydroxylates L-tryptophan to 5-hydroxy tryptophan as the first and rate-limited stage of serotonin (5-HT) synthesis in the mammalian brain [1,2]. *Tph2* gene knockout [3,4,5] or the TPH2 inhibitor p-chlorophenylalanine [6] dramatically reduce the brain’s 5-HT level. The *C1473G* mutation in the *Tph2* gene, which results in the P447R substitution in the enzyme molecule, causes a two-fold decrease in TPH2 activity in the mouse brain [7,8,9]. Taking into account the association between mutations affecting TPH2 activity and the risk of psychiatric diseases (see reviews [10,11]), it seems surprising that the *1473G* allele does not appear to produce any visible effect on adaptive behavior in laboratory mice [12,13]. However, a high (wild-type) TPH2 activity seems to be important for mouse adaptation, since mice carrying the mutant *1473G* allele, which decreases the activity of TPH2, are absent in wild mouse populations [14]. Mutations reducing the activities of aromatic amino acid hydroxylases are associated with dystonia [15]. In our preliminary study, we observed hind limb dystonia in young, 4-week-old mice of the *1473GG* genotype. However, this movement disorder disappeared in adult 12-week-old mice (see the Appendix A).

The *lethal yellow* (*A^Y^*) mutation is a large deletion in the *Raly-Agouti* locus that deletes the *Raly* gene-coding domain and puts the *Agouti* gene under the control of the *Raly* gene promoter [16,17]. It results in an ectopic expression of the agouti protein in the brain [18,19,20]. The agouti protein is a natural inhibitor of the MC3R and MC4R melanocortin receptors [21,22]. It is hypothesized that blockade of these receptors in the hypothalamus by the agouti protein causes obesity in heterozygous *A^Y^*/a mice [17,23]. Moreover, the *A^Y^* allele increased depressive-like immobility in the forced swim and tail suspension tests [24]. It can be expected that the combination of the *C1473G* and *A^Y^* alleles will increase the severity of the behavioral disorders caused by each of these mutations separately.

The aim of the present study is to compare the effects of the *C1473G* and *A^Y^* mutations in combination with the separate effects of each mutation on the behavior, brain 5-HT and melanocortin systems in mice. For this purpose, we compare the body mass, behavior, 5-HT metabolism and expression of the 5-HT-related (*Tph2*, *Maoa*, *Slc6a4*, *Htr1a*, *Htr2a*) and melanocortin-related (*Agouti*, *Mc3r*, *Mc4r*) genes in the brains of B6-1473CC/aa, B6-1473CC/*A^Y^a*, B6-1473GG/aa and B6-1473GG/*A^Y^a* mice.

## 2. Materials and Methods

### 2.1. Animals

The study was conducted in strict accordance with the recommendations of Directive 2010/63/EU of the European Parliament and of the Council of 22 September 2010 on the protection of animals used for scientific purposes and was approved by the Committee on the Ethics of Animal Experiments of the Russian National Center of Genetic Resources of Laboratory Animals of Institute of Cytology and Genetics, Siberian Branch of the Russian Academy of Sciences. The mice were bred in the Collective Centre of Animal Genetic Resources (supported by the basic research projects No. FWNR-2022-0023 and RFMEFI62117X0015). 

The experiments were carried out on SPF-state 12-week-old males of the B6-1473CC/aa (*n* = 8), B6-1473CC/*A^Y^a* (*n* = 8), B6-1473GG/aa (*n* = 8) and B6-1473GG/*A^Y^a* (*n* = 8) genotypes. The B6-1473CC/aa (50%) and B6-1473CC/*A^Y^a* (50%) genotypes were bred by crossing B6-1473CC/aa females and B6-1473CC/*A^Y^a* males. The B6-1473GG/aa and B6-1473GG/*A^Y^a* genotypes were bred in two steps. In the first step, the B6-1473GC/*A^Y^a* (50%) and B6-1473GC/aa (50%) genotypes were bred by crossing B6-1473GG/aa females and B6-1473CC/*A^Y^a* males. In the second step, the B6-1473CG/ *A^Y^a* males were crossed with B6-1473GG/aa females, and consequently, the B6-1473GG/aa (25%), B6-1473GG/*A^Y^a* (25%), B6-1473CG/aa (25%) and B6-1473CG/*A^Y^a* (25%) genotypes were received. The B6-1473CC/aa, B6-1473CC/*A^Y^a*, B6-1473GG/aa and B6-1473GG/*A^Y^a* mice used in this experiment have similar genetic backgrounds (C57BL/6) and differ only in the *1473C*, *1473G*, *a* and *A^Y^* alleles. The aa and *A^Y^*/a genotypes were detected by the color of their fur, which is yellow in *A^Y^*/a mice and black in a/a mice [17]. The *1473C* and *1473G* alleles were detected via PCR with the allele-specific primers (see Section 2.6).

After weaning, males of the same genotype were kept in groups of four per cage (Optimice, Animal Care Systems, Inc., Centennial, CO, USA) at a temperature of 24 ± 2 °C, humidity of 45–50%, and an artificial 14:10 (day/night) photoperiod with daybreak and sunset at 01:00 and 15:00, respectively. The mice were fed with sterile food and water *ad libitum*. The mice were tested at 12 weeks of age. Two days before the first test, the animals were isolated in cages of the same type and size to reduce group effect. We used a rapid test battery with 1-day intervals between tests [25]. The test sequence in these batteries was the following: open field, elevated plus maze, forced swim and tail suspension tests [26].

Two days after the tail suspension test, the animals were euthanized via carbon dioxide asphyxiation followed by decapitation. The hypothalamus, frontal cortex, hippocampus, striatum and midbrain were rapidly dissected, frozen in liquid nitrogen, and stored at −80 °C.

### 2.2. Open Field Test

The open field test was carried out on a brightly illuminated, white, opaque plastic arena 60 cm in diameter with a 30 cm wall. We used transmitted (inverted) lighting to maximize the mouse/arena contrast [27]. The light from two 12 W halogen lamps, each placed 60 cm beneath the semitransparent floor, was transmitted through the arena to a web camera. The mouse was placed at the wall, and its movements were automatically tracked for 5 min [27]. The EthoStudio software automatically calculates two behavioral traits: the distance travelled during the test (m), and the time (%) spent in the round part that is 30 cm in diameter (the center) [27]. At the same time, the numbers of vertical postures and grooming bouts were recorded by an experienced rater [27]. The arena was cleaned with wet and dry napkins after each test.

### 2.3. Elevated Plus-Maze Test

The elevated plus-maze test was carried out in an apparatus made of gray plastic with two closed and two open arms (length 30 cm and width 5 cm). The closed arms were boarded with plastic walls measuring 15 cm. The apparatus was elevated 65 cm above the floor and dimly illuminated with diffuse lighting. A mouse was placed in the apparatus center, facing a closed arm, and its movement was automatically tracked for 5 min using the Microsoft Kinect 1 3D sensor. Depth data from the sensor contain the distances (m) from each pixel of the object and each pixel of the maze to the sensor. The height threshold algorithm marks pixels higher or lower than the threshold, associated with the animal (1) or background (0), respectively [24,28]. The main advantage of the 3 D sensor over standard digital video is that the former can track an animal of any color and in any part of the maze (including the closed arms). The EthoStudio software automatically calculates three traits: the distance travelled during the test (m) and the time (%) spent in the closed and open arms, based on the rates of the number of animal-associated pixels in these parts to the total number of animal-associated pixels in the maze. The apparatus was cleaned with wet and dry napkins after each test.

### 2.4. Forced Swim Test

The mice were placed in a clear glass tank (diameter, 18 cm; height, 30 cm) filled up to 2/3 of its volume with water at a temperature of 25 °C for 6 min. The water tank was placed on a semitransparent platform and brightly illuminated (300 lx) with two halogen lamps (35 W) placed 40 cm below the platform [29]. Behavior in the FS test was evaluated for the last four minutes of the test using the immobility time (%). The water was changed after each test.

### 2.5. Tail Suspension Test

In the present study, the tail suspension test was used to measure the number and accumulated time of hind limb clasping [30]. Using adhesive tape, mice were fixed by their tails and hooked onto a horizontal bar placed 30 cm above the table surface. The number and accumulated time of hind limb clasping were registered by an experienced rater.

### 2.6. Genotyping for C1473G Polymorphism

Using protease K treatment for 1.5 h, followed by a phenol–chloroform extraction, samples of DNA were isolated from the tail tips of the mice and diluted to a concentration of 10 ng/μL. The DNA samples were genotyped according to the published protocol [7,8,9] with two positive control primers (5′-TTTGACCCAAAGACGACCTGCTTGCA and 5′-TGCATGCTTACTAGCCAACCATGAGACA) providing a 523 bp PCR product and with either *1473C* (5′-CAGAATTTCAATGCTCTGCGTGTGGG)- or *1473G* allele (5′-CAGAATTTCAATGCTXTGCGTGTGGC)-specific primers providing a 307 bp PCR product [9,10]. The PCR products were resolved via electrophoresis in 2% agarose gel, stained with ethidium bromide and photographed with a digital camera.

### 2.7. Tissue Preparation

We used the same brain samples for the total RNA extraction and for the assays of serotonin (5-HT) and 5-hydroxyindoleacetic acid (5-HIAA) levels and theTPH2 activity. For these purposes, the structure was homogenized in 250 μL (hypothalamus), 300 μL (cortex and striatum) or 400 μL (hippocampus and midbrain) of 50 mM Tris HCl, pH 7.6, and 1 mM dithiotreitol, using a motor-driven grinder (Z359971, Sigma-Aldrich, Darmstadt, Germany). One aliquot of 50 μL of the homogenate was mixed with 150 μL of 0.6 M HClO_4_ for the 5-HT and 5-HIAA extractions (see Section 2.8). Another aliquot of 100 μL of the homogenate was mixed with Trizol reagent (Bio Rad, Hercules, CA, USA) for the total RNA extraction (see Section 2.10). The rest of the homogenate was spun for 15 min at 12,700 rpm (+4 °C). The clear supernatant was transferred into a clear tube and stored at −80 °C until the TPH2 activity assay (see Section 2.9).

### 2.8. Assay of 5-HT and 5-HIAA Levels

The mix of 50 μL of homogenate with 150 μL of 0.6 M HClO_4_ (see Section 2.7) was spun for 15 min at 12,700 rpm (+4 °C). The pellet was dissolved in 1 mL of 0.1 M NaOH and used for protein determination via the Bradford method (Bio Rad, USA). The clear supernatant was diluted twofold with ultrapure water, and the levels of 5-HT and 5-HIAA were assayed in the diluted supernatant on a Luna C18(2) column (5 μm particle size, L × I.D. length 100 mm and diameter 4.6 mm, Phenomenex, Torrance, CA, USA) via electrochemical detection (750 mV, DECADE II™ Electrochemical Detector; Antec, Alphen aan den Rijn, The Netherlands), the glassy carbon flow cell (VT-03 cell 3mm GC sb; Antec, The Netherlands), a CBM-20 A system controller, the LC-20AD solvent delivery unit, a SIL-20 A autosampler and a DGU-20A5R degasser (Shimadzu Corporation, Kyoto, Japan). The mobile phase (pH = 3.2) contained 6.53 g of KH_2_PO_4_, 100 μL of 0.5M Na_2_EDTA, 150 mg of 1-octanesulfonic acid sodium salt (Sigma, Burlington, MA, USA) and methanol (13% volume; Vektor Ltd., Moscow, Russia) [24].

The standard mixes containing 1, 2 and 3 ng of 5-HT and 5-HIAA were repeatedly assayed throughout the entire procedure and used to plot the calibration curves for each substance. The peak areas were estimated using LabSolution LG/GC software, version 5.54 (Shimadzu Corporation, Kyoto, Japan), and calibrated against calibrated curves for the corresponding standards. The 5-HT and 5-HIAA contents were expressed in ng/mg protein, as assayed via the Bradford method and described elsewhere [23]. All data are the means of three replications.

### 2.9. Assay of TPH Activity

A 15 μL aliquot of pure supernatant (see Section 2.7) was incubated for 15 min at 37 °C in the presence of L-tryptophan (Sigma-Aldrich, Darmstadt, Germany) (0.4 mM), cofactor 6-methyl-5,6,7,8-tetrahydropteridine (Sigma-Aldrich, Darmstadt, Germany) (0.3 mM), decarboxylase inhibitor m-hydroxybenzylhydrazine (Sigma-Aldrich, Darmstadt, Germany) (0.3 mM), catalase (Sigma-Aldrich, Darmstadt, Germany) (5 μ) and 1 mM dithiothreitol in a final volume of 25 μL. The reaction was stopped with 75 μL 0.6 M HClO_4_ and centrifuged for 15 min at 12700 rpm. The clear supernatant was diluted twofold with ultrapure water, and the 5-HTP concentration was determined in the diluted supernatant via high performance liquid chromatography (see Section 2.8), using standards of 25, 50 and 100 pmoles of 5-HTP (Sigma, USA). Another 10 μL aliquot of supernatant was mixed with 90 μL of 0.1 M NaOH for protein quantitation via the Bradford method (Bio Rad, Hercules, CA, USA), according to the manufacturer’s protocol. The TPH activity was expressed as pmoles 5-HTP formed per minute per mg of protein, measured according to the Bradford method.

### 2.10. mRNA Level Assay by qPCR

The total mRNA was extracted from the mixture of homogenate and Trizol reagent (see Section 2.7) according to the manufacturer’s protocol and treated with RNAase-free DNAase (Promega, Madison, WI, USA) according to the manufacturer’s protocol. Its concentration was assayed with a Nanodrop 2000 (Waltham, MA, USA) and it was diluted to a final concentration of 125 ng/μL. The cDNA was synthesized using a random hexanucleotide primer and an R01 Kit (Biolabmix, Novosibirsk, Russia). The mRNA levels of the target genes were assayed via qPCR, using the set of selective primers (Table 1) and an R401 Kit (Sintol, Moscow, Russia) according to the protocol of the manufacturer (95 °C 5 min; (95 °C, 15 s; annealing temperature, 60 s; 82 °C, 2 s; fluorescence registration) × 40 cycles). The threshold cycles were calibrated with the external standards containing 25, 50, 100, 200, 400, 800, 1600, 3200 and 6400 copies of genomic DNA extracted from a C57BL/6 mouse liver. The gene expression was presented as a relative number of cDNA copies calculated on 100 copies of *Polr2* cDNA as an internal standard [24,31,32].

### 2.11. Statistics

All data were tested using Kolmogorov’s test and met the assumption of normality. Data were presented as the mean ± SEM, and they were analyzed via one-way ANOVA. If the one-way ANOVA detected an intergroup difference, the data were reanalyzed using a two-way ANOVA with “C1473G” and “*A^Y^*” as independent factors, including their interaction. Post hoc analyses were carried out using Fisher’s LSD multiple comparison test when appropriate. Statistical significance was set at *p* < 0.05.

## 3. Results

### 3.1. Effects of C1473G and A^Y^ Mutations on Body Mass

The males of the four genotypes differed in body mass (F(3,28) = 9.56, *p* < 0.001). The two-way ANOVA revealed the effects of the “C1473G” (F(1,28) = 8.9, *p* = 0.006) and “*A^Y^*” (F(1,28) = 19.69, *p* < 0.001) factors but not of their interaction (F(1,28) < 1). The *A^Y^* allele increases the body mass in B6-1473CC/*A^Y^a* and B6-1473GG/*A^Y^a* mice compared to the B6-1473CC/aa and B6-1473GG/aa mice, while the 1473G allele decreases the body mass in B6-1473GG/aa mice compared to B6-1473CC/aa mice (*p* = 0.023) (Figure 1).

### 3.2. Effects of C1473G and A^Y^ Mutations on Mouse Behavior in the Open Field Test

The males of the four genotypes differed in the distance travelled (F(3,28) = 3.04, *p* = 0.045) and the amount of fecal boli (F(3,28) = 2.95, *p* = 0.05) but not in the time spent in the center (F(3,28) < 1) or the number of vertical postures (F(3,28) = 1.54, *p* = 0.23). The two-way ANOVA revealed the effects of the “C1473G” factor (F(1,28) = 8.52, *p* = 0.007) but not of the “*A^Y^*” factor (F(1,28) < 1) or of the factors’ interaction (F(1,28) < 1) on the distance travelled: the 1473GG mice ran more distance compared to the 1473CC mice (Figure 2). Regarding a marked effect of the “*A^Y^*” factor (F(1,28) = 6.33, *p* = 0.018) but not of the “C1473G” factor (F(1,28) = 2.28, *p* = 0.14) and of their interaction (F(1,28) < 1) on the amount of fecal boli, the *A^Y^* allele increased this trait expression (Figure 2).

### 3.3. Effects of C1473G and A^Y^ Mutations on Mouse Behavior in the Elevated Plus-Maze Test

The males of the four genotypes did not differ in the distance travelled (F(3,28) = 2.2, *p* = 0.11), time spent in the center (F(3,28) = 2.02, *p* = 0.13), in open arms (F(3,28) < 1) and in closed (F(3,28) < 1) arms in the elevated plus-maze test (Figure 3).

### 3.4. Effects of C1473G and A^Y^ Mutations on Mouse Behavior in the Forced Swim Test

The males of the four genotypes differed in immobility time (F(3,28) = 4.6, *p* = 0.01) in the forced swim test. Marked effects of the “C1473G” (F(1,28) = 4.68, *p* = 0.04) and “*A^Y^*” (F(1,28) = 8.19, *p* = 0.008) factors, but not of their interaction (F(1,28) < 1), on this trait were revealed. The *C1473G* mutation decreases the immobility time, while the *A^Y^* mutation increases the immobility time (Figure 4).

### 3.5. Effects of C1473G and A^Y^ Mutations on the Hind Limb Clasping in the Tail Suspension Test

The males of the four genotypes differed in the amount of hind limb clasping (F(3,28) = 5.29, *p* = 0.005) and the accumulated time spent hind limb clasping (F(3,28) = 4.09, *p* = 0.015) in the tail suspension test. The two-way ANOVA revealed marked effects of the interaction of the “C1473G” factor (amount, F(1,28) = 8.98, *p* = 0.0023; time, F(1,28) = 8.32, *p* = 0.0031) and “C1473G” × “*A^Y^*” factors (amount, F(1,28) = 6.49, *p* = 0.008; time, F(1,28) = 3.15, *p* = 0.048) on these traits. The combination of these two mutations dramatically increases both the amount of hind limb clasping and the accumulated time spent hind limb clasping in the B6-1473GG/*A^Y^a* mice (Figure 5).

### 3.6. Effects of C1473G and A^Y^ Mutations on 5-HT, 5-HIAA Levels and the 5-HIAA/5-HT Turnover Rate in the Brain

The males of the four genotypes did not differ in 5-HT, 5-HIAA levels and the turnover rate of 5-HIAA/5-HT in the cortex (5-HT, F(3,28) = 1.86, *p* = 0.16; 5-HIAA, F(3,28) < 1; 5-HIAA/5-HT, F(3,28) < 1), hippocampus (5-HT, F(3,28) = 1.28, *p* = 0.30; 5-HIAA, F(3,28) = 1.37, *p* = 0.27; 5-HIAA/5-HT, F(3,28) < 1), striatum (5-HT, F(3,28) = 2.07, *p* = 0.13; 5-HIAA, F(3,28) = 1.67, *p* = 0.20; 5-HIAA/5-HT, F(3,28) = 1.84, *p* = 0.16), hypothalamus (5-HT, F(3,28) = 1.04, *p* = 0.39; 5-HIAA, F(3,28) < 1; 5-HIAA/5-HT, F(3,28) < 1) and midbrain (5-HT, F(3,28) = 2.26, *p* = 0.10; 5-HIAA, F(3,28) = 2.24, *p* = 0.11; 5-HIAA/5-HT, F(3,28) < 1) (Figure 6).

### 3.7. Effects of C1473G and A^Y^ Mutations on the TPH2 Activity in the Brain

The males of the four genotypes differed in TPH2 activity in cortex (F(3,28) = 23.28, *p* < 0.001), hippocampus (F(3,28) = 18.37, *p* < 0.001), striatum (F(3,28) = 33.81, *p* < 0.001), hypothalamus (F(3,28) = 11.44, *p* < 0.001) and midbrain (F(3,28) = 11.3, *p* < 0.001). The two-way ANOVA revealed a marked effect of the “C1473G” factor on the enzyme activity in the cortex (F(1,28) = 69.22, *p* < 0.001), hippocampus (F(1,28) = 54.2, *p* < 0.001), striatum (F(1,28) = 100.29, *p* < 0.001), hypothalamus (F(1,28) = 32.94, *p* < 0.001) and midbrain (F(1,28) = 32.16, *p* < 0.001). The TPH2 activity in these structures in mice with the 1473CC genotype was higher compared to mice with the 1473GG genotype (Figure 7). At the same time, no effects of the “*A^Y^*” factor (F(1,28) < 1) or the factors’ interaction (F(1,28) < 1) were shown.

### 3.8. Effects of C1473G and A^Y^ Mutations on mRNA Levels of Tph2, Maoa, Slc6a4, Htr1a, Htr2a, Agouti, Mc3r and Mc4r Genes in the Brain

The males of the four genotypes did not differ in mRNA levels of *Tph2*, *Maoa*, *Slc6a4*, *Htr1a*, *Htr2a*, *Mc3r* and *Mc4r* genes in the cortex, hippocampus, striatum, hypothalamus, and midbrain (Table 2). At the same time, significant intergroup differences in the *Agouti* genes’ mRNA levels in these structures were shown (Table 2). The two-way ANOVA revealed a significant effect of the “*A^Y^*” factor on *Agouti* gene expression in the cortex (F(1,28) = 20.54, *p* < 0.001), hippocampus (F(1,28) = 121.84, *p* < 0.001), striatum (F(1,28) = 92.79, *p* < 0.001), hypothalamus (F(1,28) = 72.11, *p* < 0.001) and midbrain (F(1,28) = 242.62, *p* < 0.001). This gene expression in these structures in mice with the *A^Y^a* genotype was higher compared to mice with the aa genotype (Table 2). At the same time, no effects of the “C1473G” factor (F(1,28) < 1) or the factors’ interaction (F(1,28) < 1) on *Agouti* gene expression in these structures were shown.

## 4. Discussion

The *A^Y^* mutation increases the body mass [17,23,24] and depressive-like immobility in the forced swim test [24,33,34]. However, this mutation did not affect the brain 5-HT system [24]. The *C1473G* mutation decreases brain TPH2 activity [8,9], but it does not seem to affect the behavior of mice in the widely used laboratory tests such as the open field, elevated plus-maze, and forced swim tests [12,13,35]. Recently this mutation was associated with an increase in hind limb dystonia in young, but not adult, B6-1473GG mice (Appendix A). It is hypothesized that the combination of the *C1473G* and *A^Y^* mutations will increase the severity of behavioral disorders caused by each of these mutations separately.

The main aim of the present study was to compare the effects of the combination of the *C1473G* and *A^Y^* mutations on body mass, behavior and the 5-HT and melanocortin systems in the brain with the effects of each of the two mutations. For this purpose, the four congenic mouse lines of the B6-1473CC/aa, B6-1473CC/*A^Y^a*, B6-1473GG/aa and B6-1473GG/*A^Y^a* genotypes were bred. It was confirmed that the *A^Y^* mutation caused the ectopic overexpression of the *Agouti* gene in the brain [20] and increased body mass and depressive-like behavior in the forced swim test [24].

A new result of this study is that the *1473G* allele decreases body mass and depressive-like immobility in the forced swim test. In other words, the *1473G* and *A^Y^* alleles have opposite effects on these traits. Moreover, no statistically significant effect of the interaction of these alleles on body mass and depressive-like immobility was shown. Therefore, the *1473G* and *A^Y^* alleles regulate these traits by means of different and independent molecular mechanisms.

The α-, β- and γ- melanocyte-stimulating hormones receptors stimulate energy expenditure and reduce eating via the hypothalamic melanocortin 3 and 4 (MC3R and MC4R) [36,37,38,39,40]. The agouti protein is a natural inhibitor of the MC3R and MC4R receptors [20,21]. Usually, this protein is not expressed in the brain; however, the *A^Y^* mutation puts the *Agouti* gene under the promoter of the ubiquitously expressing *Rally* gene [17] and causes its ectopic overexpression in the brain [18,19,20] and this study. The agouti protein inhibits the hypothalamic MC3R and MC4R receptors and reduces energy expenditure and increases eating [17].

5-HT increases eating via the 5-HT1A and 5-HT2B receptors on arcuate neurons [41]. The *Tph2* gene knockout that dramatically decreases brain 5-HT level reduces body mass in Tph2Ko mice compared to wild-type mice [4,5,42,43]. Moreover, the *Tph2* gene knockout increases the expression of the *Mc4r* gene in the hypothalamus [41], which can also reduce eating and body weight gain. Here, it was found that the *1473G* allele also reduces body mass. Although the reduction in TPH2 activity caused by the *1473G* allele is more moderate and is not accompanied with any alterations to the Mc3r and Mc4r receptors in the hypothalamus, it seems to be sufficient to reduce body weight gain.

There is an association between obesity and depressive disorders [44,45,46,47]. It can be hypothesized that obesity is the cause of the increase in depressive-like immobility observed in the forced swim test in A^Y^/a mice [24]. Indeed, some authors report that the obesity caused by the leptin receptor gene (*Lepr*) knockout increases depressive-like immobility in Lepr^db^/Lepr^db^ mice [48,49]. However, a relationship between obesity and depression seems to be more complex, and other authors do not show any increase in depressive-like behavior in these mice [50].

The relationship between TPH2 activity and depressive-like behavior is obscure. There are two commonly used indexes positively correlating to depressive-like behavior: the duration of immobility in the forced swim and tail suspension tests [51]. The effects of the TPH2 deficiency on these indexes are rather contradictory [12]: the *Tph2* gene knockout can both increase [52] and decrease [4] the immobility time, and the *1473G* allele can both decrease [9] and have no effect on [13] the immobility time in the forced swim test. The *G1449A* mutation in the *Tph2* gene, which reduces the TPH2 activity by 80%, increases immobility time in the tail suspension test [53,54,55]. However, other authors did not report this trait alteration in *Tph2* gene knockout mice [52] and in mice homozygous for the *1473G* allele [35]. In the present study, the *1473G* allele not only decreases immobility time in the forced swim test but also attenuates the increase in this trait caused by the *A^Y^* mutation.

Dystonia is a symptom of the tyrosine hydroxylase deficiency. Hind limb clasping in mice suspended by the tail is a model of dystonia [30]. Hind limb clasping was recently also found in 3–4-week-old B6-1473GG mice, while adult mice of this genotype do not show hind limb dystonia (Appendix A). The most intriguing finding in the present study is the statistically significant effect of the “C1473G” and “*A^Y^*” factors’ interaction on the hind limb clasping rate and duration. Although the *1473G* and *A^Y^* alleles separately did not show hind limb dystonia, the combination of these combination resulted in a high level of expression of this trait in adult B6-1473GG/*A^Y^a* mice. Therefore, the *A^Y^* mutation can aggravate the negative effect (dystonia) of the C1473G mutation. At the present time, the molecular mechanism of the interaction of these two mutations unknown: no alterations in the brain 5-HT system distinguishing the B6-1473GG/*A^Y^a* and B6-1473GG/aa mice have been found. As previously mentioned, possible causes of hind limb dystonia could be deficiencies in tyrosine hydroxylase or its natural cofactor, 5,6,7,8-tetrahydrobiopterin [30]. Therefore, alterations in the tyrosine hydroxylase activity and/or 5,6,7,8-tetrahydrobiopterin availability may be expected in mice heterozygous for the *A^Y^* allele. However, this hypothesis requires additional experimental verification.

The *1473G* allele is eliminated from wild mouse populations by the forces of natural selection [14]. The present study provides one hypothetical mechanism for this elimination. Although the *1473G* allele in the SPF conditions does not alter physiological functions and behavior, in extremal conditions or in combinations with other adverse mutations (for example *A^Y^*), this allele causes hind limb dystonia that can decrease mouse adaptation in wild populations.

## 5. Conclusions

The present work is a logical development of studies concerning the effects of the *A^Y^* [24] and *C1473G* [11] mutations in the *Raly-Agouti* locus and the *Tph2* gene, respectively, on the mouse brain and behavior. Here, the effects of the combination of these mutations on body mass, behavior and the brain 5-HT and melanocortin systems in B6-1473GG/*A^Y^a* mice were compared to the effects of each of these mutations separately in B6-1473CC/aa, B6-1473CC/*A^Y^a* and B6-1473GG/aa mice. The following results were obtained:As expected, the *A^Y^* allele increased body mass and depressive-like immobility in the forced swim test.The *1473G* allele decreased body mass and depressive-like immobility and increased locomotor activity in the open field test.No effect of the *A^Y^* allele on the 5-HT system and no effect of the *1473G* allele on the melanocortin system in the brain were observed.A marked effect of the combination of the *A^Y^* and *1473G* alleles on hind limb dystonia was revealed. Although these alleles do not cause hind limb dystonia separately, their combination results in a high level of expression of this trait in adult B6-1473GG/*A^Y^a* mice.

This result is the first evidence for the interaction between the C1473G and *A^Y^* mutations.

## Figures and Tables

**Figure 1 biomolecules-13-00963-f001:**
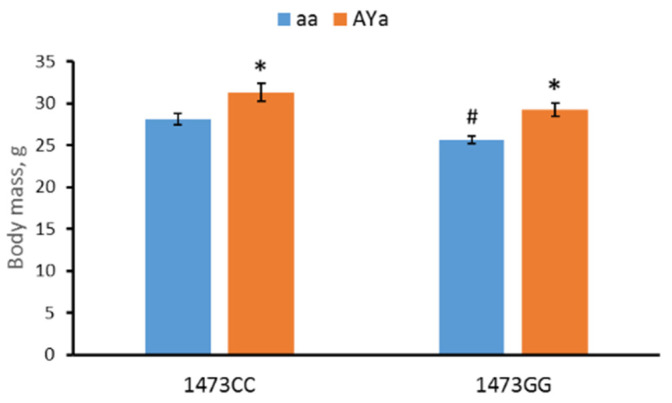
Body masses (g) of 12-week-old males of B6-1473CC/aa, B6-1473CC/*A^Y^a*, B6-1473GG/aa and B6-1473GG/*A^Y^a* mice. * *p* < 0.05 vs. corresponding aa genotype. # *p* < 0.05 vs. B6-1473CC/aa mice.

**Figure 2 biomolecules-13-00963-f002:**
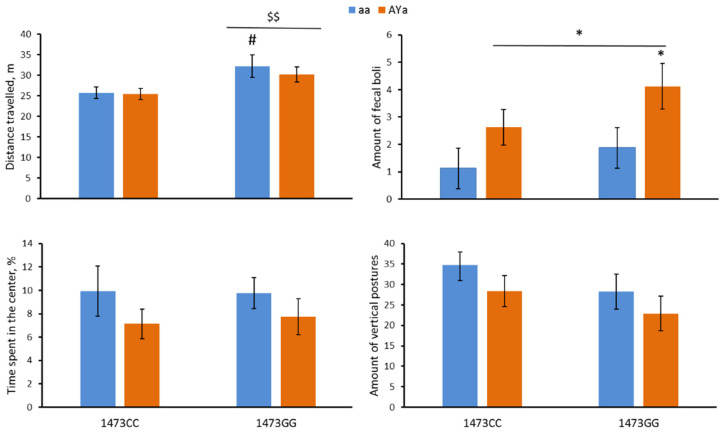
Distance travelled (m), amount of fecal boli, time spent in the center (%) and number of vertical postures in the open field test in 12-week-old male B6-1473CC/aa, B6-1473CC/*A^Y^a*, B6-1473GG/aa and B6-1473GG/*A^Y^a* mice. * *p* < 0.05 vs. corresponding aa genotype; $$ *p* < 0.01 vs. the 1473CC mice; # *p* < 0.05 vs. the B6-1473CC/aa mice.

**Figure 3 biomolecules-13-00963-f003:**
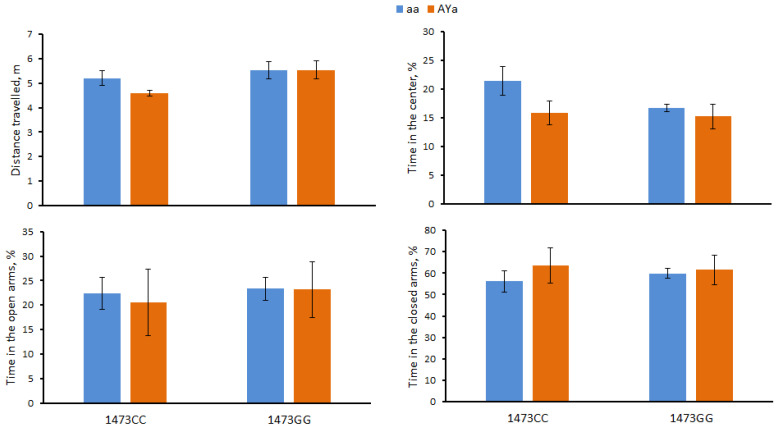
Distance travelled (m) and time spent (%) in the center, open and closed arms in the elevated plus-maze test in 12-week-old male B6-1473CC/aa, B6-1473CC/*A^Y^a*, B6-1473GG/aa and B6-1473GG/*A^Y^a* mice.

**Figure 4 biomolecules-13-00963-f004:**
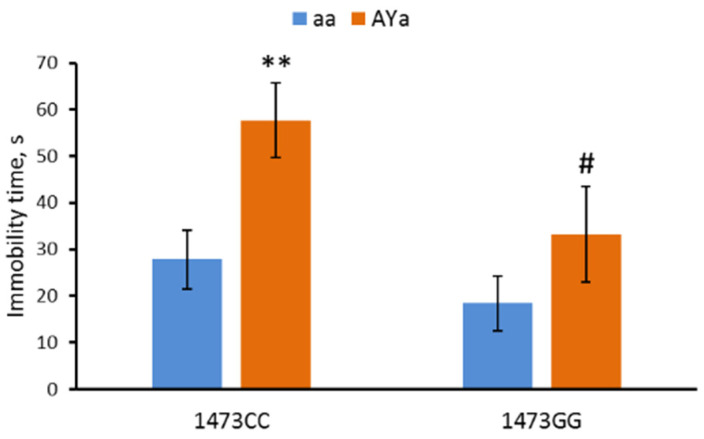
Immobility time in the forced swim test in 12-week-old male B6-1473CC/aa, B6-1473CC/*A^Y^a*, B6-1473GG/aa and B6-1473GG/*A^Y^a* mice. ** *p* < 0.01 vs. corresponding aa genotype; # *p* < 0.05 vs. the B6-1473CC/*A^Y^a* mice.

**Figure 5 biomolecules-13-00963-f005:**
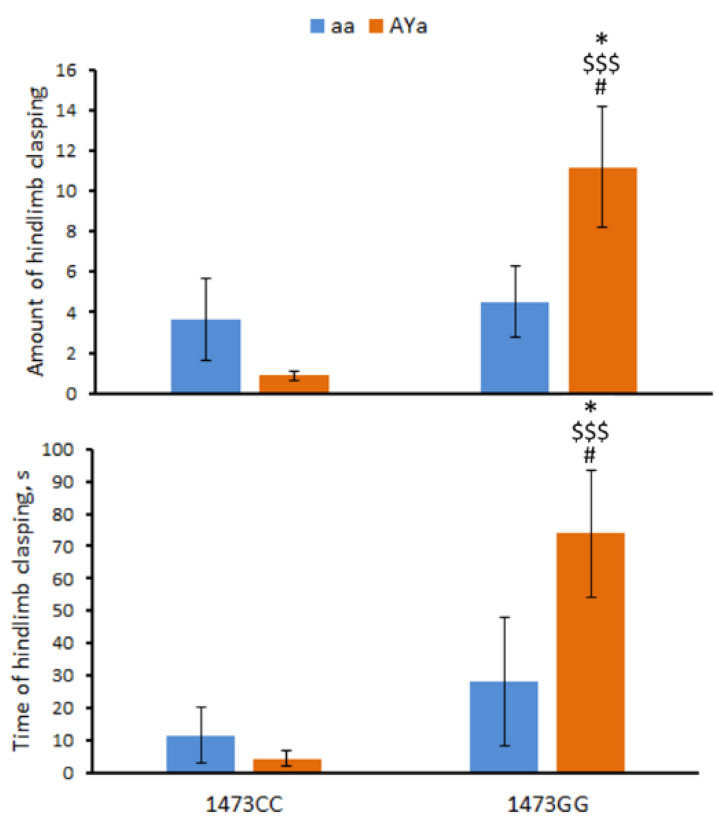
Amount of hind limb clasping and time spent hindlimb clasping in the tail suspension test in 12-week-old male B6-1473CC/aa, B6-1473CC/*A^Y^a*, B6-1473GG/aa and B6-1473GG/*A^Y^a* mice. * *p* < 0.05 vs. B6-1473CC/aa mice; $$$ *p* < 0.001 vs. B6-1473CC/*A^Y^a* mice; # *p* < 0.05 vs. the B6-1473GG/aa mice.

**Figure 6 biomolecules-13-00963-f006:**
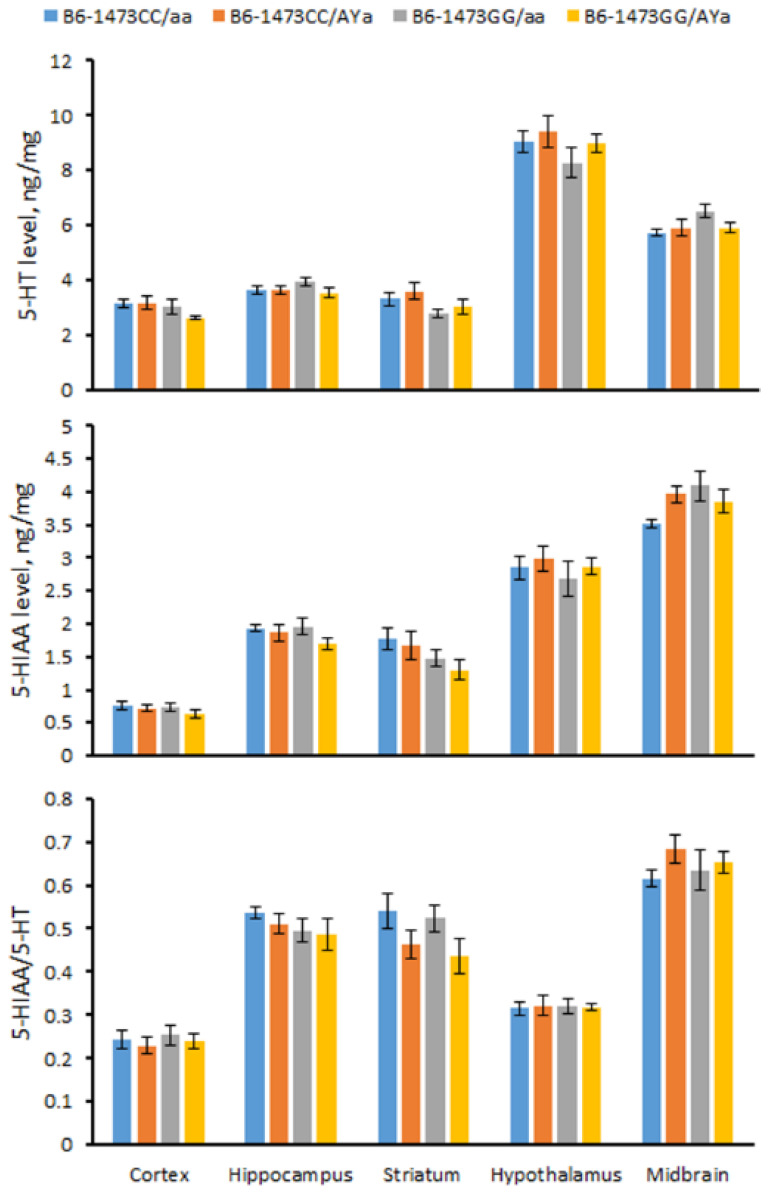
5-HT and 5-HIAA levels and the 5-HIAA.5-HT turnover rate in cortex, hippocampus, striatum, hypothalamus and midbrain in 12-week-old male B6-1473CC/aa, B6-1473CC/*A^Y^a*, B6-1473GG/aa and B6-1473GG/*A^Y^a* mice.

**Figure 7 biomolecules-13-00963-f007:**
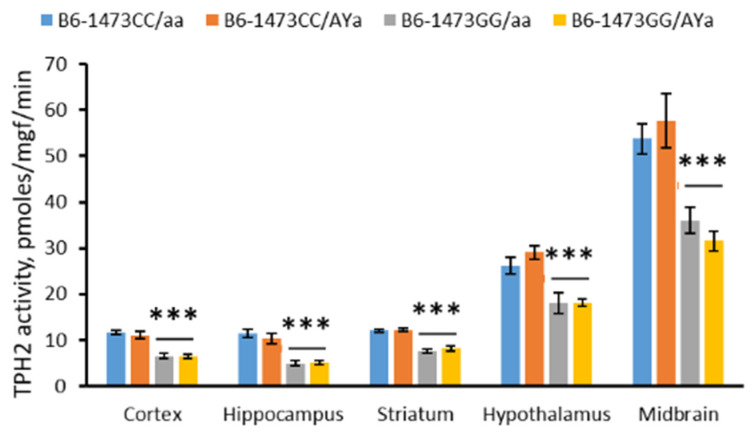
The TPH2 activity in cortex, hippocampus, striatum, hypothalamus and midbrain in 12-week-old male B6-1473CC/aa, B6-1473CC/*A^Y^a*, B6-1473GG/aa and B6-1473GG/*A^Y^a* mice. *** *p* < 0.001 vs. mice of 1473CC genotype.

**Table 1 biomolecules-13-00963-t001:** Sequences, annealing temperatures of the primers and sizes of PCR products (amplicons).

Gene	Primer Sequences	Annealing Temperature, °C	Amplicon Size, bp
** *Polr2a* **	5′-TGTGACAACTCCATACAATGC 5′-CTCTCTTACTGAATTTGCGTACT	60	194
** *Tph2* **	5′-CATTCCTCGCACAATTCCAGTCG′ 5′-AGTCTACATCCATCCCAACTGCTG	61	239
** *Maoa* **	5′AATGAGGATGTTAAATGGGTAGATGTTGGT 5′-CTTGACATATTCAACTAGACGCTC	62	138
** *Slc6a4* **	5′-AAGCCCCACCTTGACTCCTCC 5′-CTCCTTCCTCTCCTCACATATCC	57	198
** *Htr1a* **	5′-GACTGCCACCCTCTGCCCTATATC 5′-TCAGCAAGGCAAACAATTCCAG	62	200
** *Htr2a* **	5′-AGAAGCCACCTTGTGTGTGA 5′-TTGCTCATTGCTGATGGACT	61	169
** *Agouti* **	5’-GGATGTCACCCGCCTACT 5’-GTTACTCCGCAGACTCCT	62	150
** *Mc3r* **	5′-TCCGATGCTGCCTAACCT 5′-TGCAGGTTGCCATTCCT	62	130
** *Mc4r* **	5′-GTCGGAAACCATCGTCATTACC 5′-GCAAATGGATGCGAGCAAG	62	150

**Table 2 biomolecules-13-00963-t002:** The *Tph2*, *Maoa*, *Slc6a*, *Htr1a*, *Htr2a*, *Agouti*, *Mc3r* and *Mc4r* genes mRNA levels in cortex, hippocampus, striatum, hypothalamus and midbrain in 12-week-old male B6-1473CC/aa, B6-1473CC/*A^Y^a*, B6-1473GG/aa and B6-1473GG/*A^Y^a* mice.

Gene	B6-1473CC/aa	B6-1473CC/*A^Y^a*	B6-1473CC/aa	B6-1473CC/*A^Y^a*	F(3,28)
Cortex					
*Maoa*	650.3 ± 128.3	556.2 ± 50.1	632.2 ± 82.8	602.0 ± 52.4	F < 1
*Htr1a*	34.0 ± 6.6	33.5 ± 4.6	37.0 ± 3.7	57.0 ± 19.2	F = 1.2, *p* = 0.3
*Htr2a*	126.0 ± 16.1	119.6 ± 11.0	140.5 ± 23.5	138.5 ± 11.4	F < 1
*Agouti*	0.56 ± 0.42	36.3 ± 11.5 ***	0.07 ± 0.03	24.1 ± 6.4 **	F = 7.4, *p* < 0.001
*Mc3r*	0.72 ± 0.32	0.46 ± 0.09	0.52 ± 0.09	1.14 ± 0.47	F = 1.1, *p* = 0.4
*Mc4r*	12.20 ± 2.57	8.81 ± 1.37	11.25 ± 2.29	9.09 ± 1.54	F < 1
Hippocampus					
*Maoa*	200.0 ± 20.6	185.4 ± 17.0	185.6 ± 18.0	187.0 ± 23.6	F < 1
*Htr1a*	53.5 ± 7.0	45.2 ± 4.6	40.1 ± 6.4	59.9 ± 7.2	F < 1
*Htr2a*	10.5 ± 0.86	10.3 ± 1.30	10.3 ± 2.17	11.5 ± 1.59	F < 1
*Agouti*	1.25 ± 0.56	30.8 ± 3.3 ***	2.23 ± 1.95	31.3 ± 3.7 ***	F = 40.6, *p* < 0.001
*Mc3r*	3.58 ± 1.33	1.68 ± 1.09	1.10 ± 0.54	3.31 ± 1.87	F < 1
*Mc4r*	17.29 ± 2.58	14.84 ± 2.27	18.83 ± 7.06	16.39 ± 2.61	F < 1
Striatum					
*Maoa*	106.1 ± 9.9	114.5 ± 8.8	120.4 ± 13.9	127.2 ± 13.8	F < 1
*Htr1a*	2.61 ± 0.57	3.79 ± 0.53	3.82 ± 0.54	3.51 ± 0.45	F = 1.2, *p* = 0.3
*Htr2a*	28.7 ± 2.54	30.5 ± 5.62	25.2 ± 3.35	32.5 ± 3.61	F < 1
*Agouti*	0.14 ± 0.12	42.1 ± 7.6 ***	0.03 ± 0.01	61.1 ± 7.5 ***	F = 33.0, *p* < 0.001
*Mc3r*	0.50 ± 0.14	1.46 ± 0.49	0.66 ± 0.10	1.06 ± 0.29	F = 2.1, *p* = 0.1
*Mc4r*	6.90 ± 1.15	6.61 ± 1.07	6.91 ± 1.12	7.36 ± 1.05	F < 1
Hypothalamus					
*Maoa*	242.7 ± 21.3	228.7 ± 18.9	249.9 ± 16.8	294.3 ± 46.4	F < 1
*Htr1a*	22.8 ± 1.6	20.9 ± 2.2	20.4 ± 2.3	24.8 ± 4.7	F < 1
*Htr2a*	16.81 ± 1.08	18.74 ± 2.43	18.94 ± 1.86	21.88 ± 4.36	F < 1
*Agouti*	0.21 ± 0.19	40.5 ± 4.2 ***	0.02 ± 0.01	45.6 ± 9.2 ***	F = 24.2, *p* = 0.001
*Mc3r*	12.41 ± 1.18	17.67 ± 7.48	13.54 ± 2.31	13.91 ± 3.94	F < 1
*Mc4r*	10.55 ± 0.62	9.91 ± 1.72	10.29 ± 1.09	11.63 ± 2.48	F < 1
Midbrain					
*Tph2*	337.2 ± 59.1	261.2 ± 51.3	375.1 ± 19.2	283.6 ± 35.8	F < 1
*Maoa*	811.73.1	809.9 ± 70.6	847.9 ± 86.0	787.8 ± 50.0	F < 1
*Slc6a4*	78.4 ± 9.9	57.8 ± 4.7	64.3 ± 6.9	81.0 ± 19.2	F < 1
*Htr1a*	40.7 ± 3.6	34.2 ± 2.5	30.8 ± 2.8	34.4 ± 2.7	F = 1.8, *p* = 0.2
*Htr2a*	36.0 ± 3.84	30.8 ± 2.32	29.3 ± 2.45	40.9 ± 9.73	F < 1
*Agouti*	5.54 ± 1.24	46.4 ± 2.4 ***	3.45 ± 0.83	48.2 ± 4.7 ***	F = 81.0, *p* = 0.001
*Mc3r*	10.21 ± 1.61	3.93 ± 1.43	6.42 ± 1.65	6.49 ± 3.06	F = 2.0, *p* = 0.1
*Mc4r*	20.13 ± 4.97	16.38 ± 2.51	15.36 ± 2.33	15.60 ± 3.19	F < 1

** *p* < 0.01, *** *p* < 0.001 vs. corresponding aa genotype.

## Data Availability

Not applicable.

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
