# Peer review of "Effects of the Combination of the C1473G Mutation in the Tph2 Gene and Lethal Yellow Mutations in the Raly-Agouti Locus on Behavior, Brain 5-HT and Melanocortin Systems in Mice"

_biomolecules, 2023, doi:10.3390/biom13060963_

Round 1
Reviewer 1 Report
The study by Komleva et. Al. addresses a possible interaction between a TPH2 mutation and the ectopic expression of Agouti. The rational of this study needs clarifying in that no known molecular interaction of these systems has been identified and the reason to study the combination of those 2 genetic defects is not clearly stated. Most of the results presented show no statistical interaction of those 2 mutations in most of the phenotypes measured except for dystonia. Without any molecular mechanism insight, this study is correlative and does not demonstrate interaction between the serotonin and melanocortin system. It would be helpful to have a description of the questions that each experiment aims to answer in each section of the result section. In the current format, it is a bit difficult to follow the logical progression. What are the n values for each experiment?
To increase the significance of this study, more experiments are needed:
1) Is the weight difference between the Ay/1473CC and Ay/1473GG mice contributing to the increased dystonia? Would pair fed animals show the same phenotype?
2) Is this difference due to MC4R inhibition? Would MC4R KO/ 1473GG present the same phenotype?
3) Same question for MC3R
Author Response
Reviewer 1
The study by Komleva et. Al. addresses a possible interaction between a TPH2 mutation and the ectopic expression of Agouti. The rational of this study needs clarifying in that no known molecular interaction of these systems has been identified and the reason to study the combination of those 2 genetic defects is not clearly stated. Most of the results presented show no statistical interaction of those 2 mutations in most of the phenotypes measured except for dystonia. Without any molecular mechanism insight, this study is correlative and does not demonstrate interaction between the serotonin and melanocortin system. It would be helpful to have a description of the questions that each experiment aims to answer in each section of the result section. In the current format, it is a bit difficult to follow the logical progression. What are the n values for each experiment?
To increase the significance of this study, more experiments are needed:
1) Is the weight difference between the Ay/1473CC and Ay/1473GG mice contributing to the increased dystonia? Would pair fed animals show the same phenotype?
2) Is this difference due to MC4R inhibition? Would MC4R KO/ 1473GG present the same phenotype?
3) Same question for MC3R
Answer: Dear colleague, I’m sorry, that I failed to introduce you the main idea of our study. It’s my fault and I beg your pardon. Our task was only to investigate the interaction between the C1473G and AY mutations, but not between the 5-HT and melanocortin systems. You are quite right that the second task requires a completely different experimental design. And it will other study. So, I’ve corrected the manuscript text in order to avoid misunderstanding.
I thank you very much for helping to identify and correct this misunderstanding.
Reviewer 2 Report
Polina Komleva, Ghofran Alhalabi, Arseniy Izyurov, Nikita Khotskin, Alexander Kulikov
Effects of combination of C1473G mutation in Tph2 gene and lethal yellow mutations in the Raly-Agouti locus on behavior, brain 5-HT and melanocortin systems in mice.
COMMENTS FOR THE AUTHOR:
1. The presented research is an original and important for physiology and neurobiology. The manuscript is included all parts which needs for the publication: Abstract, Introduction, Materials and Methods, Results, Discussion, Conclusion, References.
2. The title clearly and precisely reflects the findings of the manuscript.
3. Abstract is it really a summary, include key findings and have an appropriate length.
4. This study and its introduction, in particular, give an idea of which mechanisms at the gene level are involved in the formation of behavioral functions and are important for the survival of an individual. We are talking about the interaction of different genes, in this study, first of all, the serotonin and melanocortin systems are considered.
5. The authors described the methods in detail, they are sufficient for reproduction by other researchers. There is no need for additional materials.
6. The general logic of the results is correct, the pictures suggested and located strictly in accordance with the sсheme of section. In my opinion, additional experiments are not required.
7. Received results indicate that the 5-HT and melanocortin system can interact in the mechanisms of behavioral disorders and B6-1473GG/AYa mice is a promising model for experimental investigation of the molecular mechanisms underlying this interaction.
8. The figures correspond to the article’s structure; legends to him explain the drawings. The citation is appropriate, the included the basic publications on the topic.
9. Final comments.
The manuscript is fully consistent to the stated theme. I recommend for publication without revision.
Author Response
Effects of combination of C1473G mutation in Tph2 gene and lethal yellow mutations in the Raly-Agouti locus on behavior, brain 5-HT and melanocortin systems in mice.
COMMENTS FOR THE AUTHOR:
1. The presented research is an original and important for physiology and neurobiology. The manuscript is included all parts which needs for the publication: Abstract, Introduction, Materials and Methods, Results, Discussion, Conclusion, References.
2. The title clearly and precisely reflects the findings of the manuscript.
3. Abstract is it really a summary, include key findings and have an appropriate length.
4. This study and its introduction, in particular, give an idea of which mechanisms at the gene level are involved in the formation of behavioral functions and are important for the survival of an individual. We are talking about the interaction of different genes, in this study, first of all, the serotonin and melanocortin systems are considered.
- The authors described the methods in detail, they are sufficient for reproduction by other researchers. There is no need for additional materials.
- The general logic of the results is correct, the pictures suggested and located strictly in accordance with the sсheme of section. In my opinion, additional experiments are not required.
- Received results indicate that the 5-HT and melanocortin system can interact in the mechanisms of behavioral disorders and B6-1473GG/AYa mice is a promising model for experimental investigation of the molecular mechanisms underlying this interaction.
- The figures correspond to the article’s structure; legends to him explain the drawings. The citation is appropriate, the included the basic publications on the topic.
- Final comments.
The manuscript is fully consistent to the stated theme. I recommend for publication without revision.
Answer: Thank you very much for your appreciation of our work.
Reviewer 3 Report
The manuscript "Effects of combination of C1473G mutation in Tph2 gene and lethal yellow mutations in the Raly-Agouti locus on behavior, brain 5-HT and melanocortin systems in mice" by Komleva et al. discusses in detail about the interaction between the lethal yellow and C1473G mutations. According to the author's findings lethal yellow mutation increases the hind limbs dystonia caused by the 1473G allele.
Specific comments:
The abstract might benefit from some rephrasing.
Introduction: Please include little bit more details.
Methods and results are very well written.
Discussion: the possibilities and limitations of this work are missing/ not properly addressed.
Author Response
The manuscript "Effects of combination of C1473G mutation in Tph2 gene and lethal yellow mutations in the Raly-Agouti locus on behavior, brain 5-HT and melanocortin systems in mice" by Komleva et al. discusses in detail about the interaction between the lethal yellow and C1473G mutations. According to the author's findings lethal yellow mutation increases the hind limbs dystonia caused by the 1473G allele.
Specific comments:
The abstract might benefit from some rephrasing.
Answer: We have rephrased the abstract in order to improve its logical structure.
Introduction: Please include little bit more details.
Answer: We have corrected the introduction in order to improve the understanding the problem and the aim and tasks of our study. At the same time, we tried not to overload the introduction with more information than is necessary to understand the problem and objectives of the study. Therefore, we moved all additional information to the discussion.
Methods and results are very well written.
Discussion: the possibilities and limitations of this work are missing/ not properly addressed.
Answer: The discussion has been completely rewritten. New 8 references have been added and 3 old references have been removed.
We thank you very much for your valuable comments.
Round 2
Reviewer 1 Report
corrections were appropriate